# Spatio-Temporal Characteristics of Dry-Wet Conditions and Boundaries in Five Provinces of Northwest China from 1960 to 2020

**Miao Wang** **, Puxing Liu *, Xuemei Qiao, Wenyang Si and Lu Liu**

Key Laboratory of Resource Environment and Sustainable Development of Oasis,
College of Geography and Environmental Science, Northwest Normal University, Lanzhou 730070, China;
wwangmxl@163.com (M.W.); qiaoxm751228@163.com (X.Q.); nwnuswy@163.com (W.S.); liul3191@163.com (L.L.)
* Correspondence: liupx751228@163.com

**Abstract:** The study of dry-wet climate boundaries in the context of climate warming is of great practical significance for improving the environment of ecologically fragile zones and promoting economic and natural sustainable development. In this study, based on the daily meteorological data of 110 stations, using the wetness index, empirical orthogonal function decomposition, regime shift detection test, Fourier power spectrum, and Kriging interpolation, the researchers analyzed the spatiotemporal characteristics of dry-wet conditions and boundaries in five provinces of Northwest China from 1960 to 2020. The results showed that the overall wetness index increased in the past 61 years, but with significant internal differences, among which the western and central climate tended to be warm and wet, and the eastern tended to be warm and dry. The annual wetness index changed abruptly in 1986 with cycles of 3.61 a, 7.11 a and 8.83 a. The mutations occurred correspondingly in spring, summer, autumn, and winter in 1972, 1976, 1983, and 1988, with periods of 3.88 a and 4.92 a, 2.18 a and 2.81 a, 2.15 a, and 2.10 a, respectively. The dry-wet climate boundary has fluctuated markedly since 1960. The extreme arid and arid regions boundary shifted southward and shrank in size until the extreme arid region disappeared in the 2010s. The arid along with semi-arid regions and semi-arid in addition to semi-humid regions boundaries both have two boundary lines, and show the shift of the northwestern boundary to the southeast and the southeastern boundary to the northwest, with the area of the arid together with semi-arid regions shrinking significantly by 5.64%, simultaneously, the area of the semi-humid region area expanding significantly by 84.11%. The boundary of semi-humid and relatively humid regions, and the boundary of relatively humid and humid regions all shifted to the southeast, moreover, the area of the relatively humid region and humid region shrank significantly by 12.08%. The expansion of semi-humid region and the contraction of other climate regions are characteristics of the dry-wet climate variability in five provinces of Northwest China. The area of the three arid climate zones dwindled by 9.61%, and the area of the three humid zones extended by 39.01%. Obviously, the climate inclined to be warm and humid in general.

**Keywords:** dry-wet climate boundary; EOF decomposition; STARS; Fourier power spectrum; northwest region

## 1. Introduction

The abnormal climate change events caused by global warming have become an issue of great concern all over the world. It has an impact on the dry-wet climate conditions, which will lead to a change in the dry-wet climate boundaries, the fluctuation of which visually reflects the dry-wet climate variability of a region.

The concept of the wetness index was first proposed by Dukuchaev in 1900 [1], and Vysotsky [2,3] was the first to calculate quantitatively the ratio of the wetness index in 1905; there were differences in the calculation of the wetness index due to different methods of

evapotranspiration. Thornthwaite [4] put forward the P-E index to estimate the possible evapotranspiration based on temperature, the sunshine hours in 1931, and revised the P-E index in 1948 and 1955. Holdridge [5] raised the calculation of the probable evapotranspiration rate (PER) based on the precipitation and plant bio-temperature in 1947. Budyko [6] came up with the radiation drying index to calculate the probable evapotranspiration capitalizing on the heat and water balance principles in 1948. Penman [7] elaborated a formula for calculating the potential evapotranspiration under the condition of no horizontal transport of water vapor and revised it in 1956. A research group funded by the American Society of Civil Engineers (ASCE) compared the accuracy of 20 formulas for calculating the potential evapotranspiration by 11 kinds of measured data obtained under different climatic conditions around the world. The results indicated that the Penman–Monteith formula has the highest accuracy in both wet and dry regions. Furthermore, similar studies conducted by related scholars have also verified this assumption [8–10]. The Rome Workshop on methods for calculating crop water requirements also recommended the use of the Penman–Monteith formula for calculation of the reference crop evapotranspiration in 1990 [11]. The Food and Agriculture Organization of the United Nations (FAO) proposed the Penman–Monteith formula as the only standard method for the reference crop evapotranspiration calculations and renamed it the FAO Penman–Monteith formula in 1998 [12]. Subsequently, the FAO Penman–Monteith formula was extensively adopted by domestic and foreign scholars to calculate the wetness index to study the variability of dry-wet conditions and climate boundaries in different regions. Spinoni et al. [13] analyzed the transformation of global climate zones and found that the area of arid climate regions increased by nearly 1.4% from 1981 to 2000 relative to 1961–1980. Feng et al. [14] suggested that the area of global arid zones will expand by 10% at the end of the 21st century compared with 1961–1990. Gao et al. [15] scrutinized the variability of dryness and wetness on the Tibetan Plateau from 1979 to 2011, the results indicated that most of the eastern part of the plateau became dry and about half of the northwestern part became wet. Ma et al. [16] found that precipitation increased significantly in the eastern part of China in the last 100 years and the area became significantly wetter, while the western part did not show a trend of wetting despite the increase in precipitation. Wang et al. [17] found that northern China and southern northeastern China were still persistently in a dry period, but the trend of damp intensity has been observed in northern areas.

When the change in wet and dry conditions reach a certain degree, it will lead to the movement of the wet and dry climate boundary, and the fluctuation characteristics of the wet and dry boundary are significantly different in different regions. In the past 50 a, the wet and dry climate boundary in China has been moving as a whole and fluctuating from east to west and from north to south [18]; it mainly shows the contraction of the semi-humid zone and the expansion of the semi-arid zone [19]; while another study found that although China tends to be arid as a whole, the future wet and dry evolution is characterized by the contraction of the humid zone, arid zone and extreme arid zone, and the expansion of the semi-humid zone and semi-arid zone [20]; the wet and dry boundary of the Qinghai–Tibet Plateau moved northwest, the extreme arid zone and arid zone contracted, and the semi-arid zone, semi-humid zone and humid zone expanded [21]; the dry and wet boundary in Shanxi Province moved northwest and then southeast, and the area of arid zone expanded and then contracted [22]; the semi-arid and humid zone and the warm temperate zone and subtropical boundary in the Qinling region shifted northward in the past 30 years, and the trend of climate warming and drying was obvious [23]; while the upper reaches of the Yellow River [24], Northeast China [25], and the Loess Plateau [26] show that the dry-wet boundary shifts, the semi-arid zone expands, the semi-humid zone contracts, and the climate tends to warm and dry. A causal analysis indicated that the southeast monsoon and southwest monsoon and westerly circulation caused by the western Pacific subtropical high pressure and the Bay of Bengal warm current are the main causes of the fluctuation of the climate boundary in China [27,28]; while the fluctuation of the wet and dry boundary in Liaoning Province is controlled by the western Pacific subtropical high

pressure, southeast monsoon, and topographic precipitation [29]; the fluctuation of the climate boundary in northern China mainly depends on the change in precipitation and potential evaporation rate [30]; the fluctuation of the wet and dry climate boundary is controlled only by natural factors, but the climate boundary of the agricultural-pastoral transition zone is also controlled by human factors [31].

Most of five provinces of Northwest China is located in the arid and semi-arid regions of China, which is a sensitive area for climate change and a fragile ecological environment. At the end of last century, Shi et al. [32] concluded that the climate of the Five Northwestern Provinces underwent a shift from warm-dry to warm-wet around 1987 and some scholars have also conducted a lot of studies on the dry-wet climate variability in five provinces of Northwest China [33–35]. What are the spatial and temporal characteristics of the dry and wet climate boundaries in the northwest? What are the characteristics of dry and wet climate evolution? Is it consistent with the conclusions drawn from existing studies [18–20]? Therefore, based on the daily meteorological data from 1960 to 2020, this study uses the internationally unified the wetness index to reveal the spatial and temporal changes in dry and wet conditions and dry and wet boundaries in five provinces of Northwest China over the past 61 years, with a view to providing a scientific decision basis for improving the ecological environment, rationalizing the layout of agricultural production, and promoting high-quality and stable development in five provinces of Northwest China.

## 2. Materials and Methods

### 2.1. Study Area

The Five Northwestern Provinces is located between 31°32′ and 49°10′ N, 73°15′ and 111°15′ E, an is comprised of five provinces, Gansu Province, Qinghai Province, Shaanxi Province, Xinjiang Uyghur Autonomous Region, and Ningxia Hui Autonomous Region, covering an area of about 3.043 million km$^2$, accounting for 31.7% of the national land area (Figure 1). The area is located in the first and second echelons of China and has the distinction of alternative-distribution of a plateau, mountain, basin, and plain. The climate type is complex and diversified, with the overall climatic attributes of aridity and low rainfall, strong evaporation, long sunshine hours and uneven precipitation distribution and the average annual temperature range from −11.96 to 17.61 °C, the sunshine hours aggregate 1433.71 to 3507.86 h, the annual mean precipitation is between 8.24 and 1503.16 mm with a decrease from southeast to northwest. The main types of soil contain brown calcium soil, gray calcium soil, desert soil, black kiln soil, yellow cotton soil, etc. The types of vegetation are complicated and diverse, mainly composed of subtropical broad-leaved forests, cold coniferous forests, alpine meadows, and desert vegetation such as poplar, tamarisk, and sorrel.

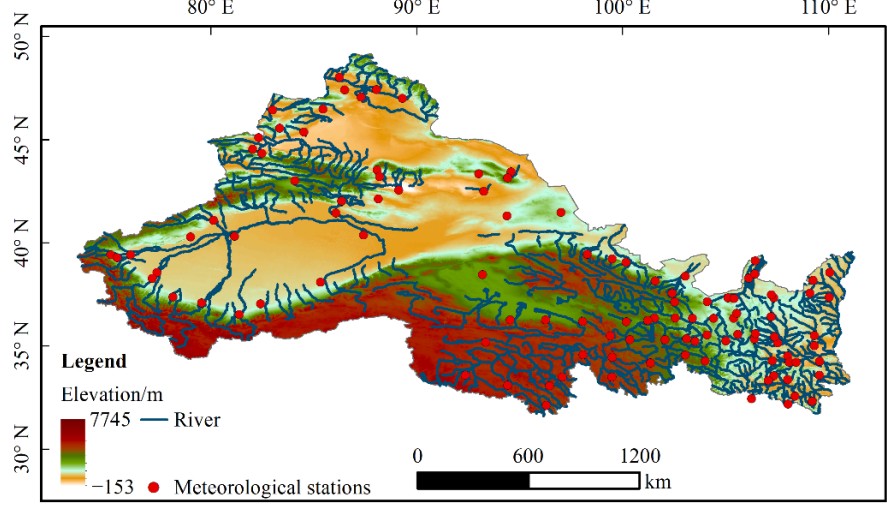

**Figure 1.** Distribution of meteorological stations in five provinces of Northwest China.

*2.2. Data Resources and Methods*

2.2.1. Data Sources

In this study, after eliminating the stations with more dissipated measurements and stations relocated and withdrawn, we selected the daily meteorological data of the maximum temperature, minimum temperature, average temperature, average wind speed, sunshine hours, relative humidity, precipitation, etc., extracted from 110 meteorological observation stations in five provinces of Northwest China from 1960 to 2020 and the data were obtained from China Meteorological Data Network (https://data.cma.cn) (accessed on 3 April 2021).

2.2.2. Methods

(1) Calculation method of the wetness index

In this paper, the wetness index (W) is used as a criterion for the division of dry-wet climate zones and its calculation formula is as follows [25]:

$$W = \frac{P}{ET_0} \tag{1}$$

where P is the daily precipitation (mm) and $ET_0$ is the daily reference crop evapotranspiration (mm). The greater the W, the wetter the climate, and vice versa.

The reference crop evapotranspiration (ET0) is calculated by the Penman–Monteith formula recommended by the Food and Agriculture Organization of the United Nations (FAO) in 1998. The equation is:

$$ET_0 = \frac{0.048\Delta(R_n - G) + \gamma\frac{900}{T+273}u_2(e_s - e_a)}{\Delta + \gamma(1 + 0.34u_2)} \tag{2}$$

where $R_n$ is the daily net radiation (MJ/(m2-d)); G is the soil heat flux (MJ/(m2-d)); $\gamma$ is psychrometric parameter (kPa/°C); T is the mean air temperature (°C); $u_2$ is the wind speed at a height of 2 m from the ground (m/s); $e_s$ is the atmospheric saturated water pressure (kPa); $e_a$ is the actual atmospheric water pressure (kPa); $\Delta$, the change in air pressure with temperature, is the slope of the curve of the relationship between air pressure and temperature (kPa/°C). See the literature for details of the calculation [36].

The classification of dry-wet conditions generally adopts the climatic dry-wet zones classification criteria established by the United Nations for the universal desertification problem [37] and in this study, we choose the climatic dry-wet zones classification criteria (Table 1) improved and refined by Wang et al. [30] on this basis in order to better highlight the evolution of dry-wet conditions in the northwest.

**Table 1.** Grades of wetness index based on Chinese climate classification criterion.

|  | **Dry-Wet Climate Zones** |
|---|---|
| <0.03 | Extreme arid region |
| 0.05–0.20 | Arid region |
| 0.20–0.50 | Semi-arid region |
| 0.50–0.75 | Semi-humid region |
| 0.75–1.0 | the relatively humid region |
| >1 | Humid region |

(2) Empirical Orthogonal Function Decomposition

The empirical orthogonal function decomposition was first proposed by Pearson [38] and is also known as principal component analysis. The basic principle is to decompose the variable field into a spatial field and a temporal field. The eigenvectors that pass the significance test can maximize the distribution structure of the climate variable field in a region. They represent the spatial distribution type as the most typical distribution structure of the variable field, while the eigenvectors of positive and negative interphase

distribution represent two types of distribution. The temporal coefficients embody the time-varying characteristics of the distribution type represented by the eigenvectors. The positive and negative coefficients are on behalf of the positive and negative distributions of the distance level of the variable. The calculation procedure is described in reference [39].

(3) Regime Shift Detection

The regime shift detection test, also known as the sequential-test analysis of regime shifts (STARS), was first advanced by Sergei N. Rodionov in 2004 [40]. STARS can visually detect the timing of pattern shifts and has been widely applied in studies on climate change, for instance. In this study, we use the regime shift detection method through Matlab to test for abrupt changes in dry-wet changes in five provinces of Northwest China from 1960 to 2020. The detailed calculation procedure is described in reference [40].

(4) Fourier power spectrum analysis.

In this paper, using Matlab to analyze the period of the wetness index in five provinces of Northwest China, and for a given discrete time series $X_0$, $X_1$, ... , $X_{n-1}$, the Fourier transform of the discrete-time series $\{X_n\}$ can be expressed as:

$$C_k = F(x_n) = \sum_{n=0}^{N-1} X_n e^{-j\frac{2\Pi}{N}kn} \tag{3}$$

where $k = 0, 1, \ldots, N - 1$, $C_k$ is the Fourier transform coefficient of the discrete time series $\{X_n\}$, which can be divided into amplitude $|C_k|$ and phase $\tan^{-1}[\alpha\{C_k\}/\beta\{C_k\}]$ real part $\alpha\{C_k\}$ and imaginary part $\beta\{C_k\}$, or real part $\alpha\{C_k\}$ and imaginary part $\beta\{C_k\}$. Therefore, we define $|c_k|^2$ as the Fourier power spectrum of the signal [39].

## 3. Results

### 3.1. Spatial Distribution Characteristics of the Wetness Index

3.1.1. Annual Spatial Distribution Characteristics

In this study, the empirical orthogonal function (EOF) combined with the North test are used to decompose the wetness index of 110 meteorological stations in five provinces of Northwest China from 1960 to 2020 [38], ArcGIS and Origin are deployed to plot the spatial distribution of the EOF decomposition and the corresponding temporal weight coefficients (Figure 2). The results show that variance contributions of the first three modes were 30.31%, 24.05%, and 12.69%, respectively and the cumulative contribution is 67.05% (Table 2), which passed the North significance test. Therefore, the first three modes can better reflect the spatial distribution characteristics of the wetness index in the northwest.

As mentioned above, Table 2 and Figure 2 EOF1 manifest that the first mode has the highest variance contribution of 30.31%, indicating the first mode being primary the main form of the spatial distribution of the wetness index in five provinces of Northwest China where positive regions are mainly distributed in Xinjiang, northwestern Qinghai, Jiuquan in Gansu, and some parts of Wuwei City, and in addition to other negative regions, implying an opposite distribution type of the wetness index changes with Xinjiang and other places. Combined with the change curve of the first modal time coefficient (Figure 2 PC1), the linear trend of the time coefficient in the northwest region has escalated in the past 61 years, with an overall trend of becoming wetter. The inflection point of the positive and negative values was in 1976, indicating that before 1976, except for Xinjiang and other places where the wetness index was a negative pitch level, all other regions had a positive pitch level, while since 1976 in Xinjiang and other places a negative pitch level changed to a positive pitch level. Later, in Xinjiang and other areas the wetness index changed from negative to positive, showing an increasing trend, while in other areas the wetness index showed a sharp contrast with that of Xinjiang. The above results are consistent with the conclusion that the northwest region as a whole is warmer and wetter, but local differences show that the eastern part of the northwest began to incline to be warmer and drier at the end of the 1970s, and the western part tended to be warmer and wetter [35,41].

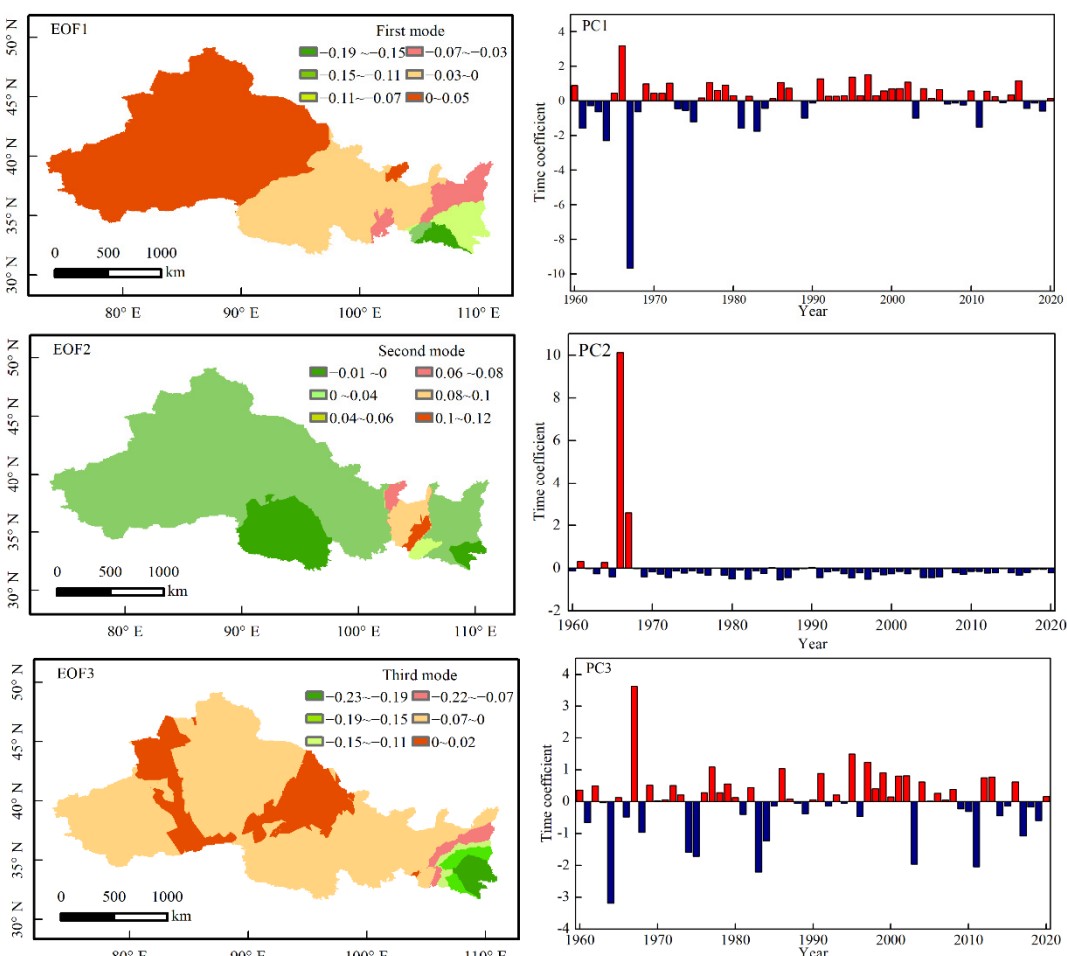

**Figure 2.** Spatial distribution (**the left**) and time coefficients (**the right**) of first three modes of EOF analysis of humidity index in five provinces of Northwest China.

**Table 2.** Statistical results for the first 5 modalities obtained by empirical orthogonal function (EOF).

| Mode | 1 | 2 | 3 | 4 | 5 |
|---|---|---|---|---|---|
| Variance contribution (%) | 30.31 | 24.05 | 12.69 | 7.08 | 6.14 |
| Accumulated variance (%) | 30.31 | 54.36 | 67.05 | 74.13 | 80.27 |

The second mode (Figure 2 EOF2) shows a negative area in spatial distribution except for southern Qinghai, southern Shaanxi Ankang, and Shangluo and a positive area in the rest of the regions. From the PC2 plot, it can be obtained that the time coefficients are all negative after 1968, indicating that the wetness index in southern Qinghai, Ankang, and Shangluo has an intensive trend after 1968, while the rest of the regions have a dissipating trend.

The spatial type of the third mode (Figure 2 EOF3) shows the inverse change characteristic of eastern Xinjiang and the central belt region and other regions. The positive and negative inflection point of the time coefficients is 2008, indicating that the wetness index tends to decrease in eastern Xinjiang and the central belt region from 1960 to 2008; however, after 2008, the wetness index tends to increase and the climate tends to be warm and wet, by contrast, the opposite is true for other regions.

3.1.2. Spatial Distribution Characteristics of the Trend of the Wetness Index

In the past 61 years, there has been a significant spatial distribution difference in the interannual distribution of the tendency rate of the wetness index change in five provinces

of Northwest China (Figure 3), with the tendency rate gradually decreasing from west to east, i.e., the tendency rate in the western and central regions is mainly positive, accounting for 88.24%, and the wetness index is increasing, on the contrary, while in the eastern region the tendency rate is mainly negative, accounting for 11.76%, and the wetness index is decreasing, which is consistent with the spatial distribution of the wetness index obtained in the previous study. The areas with the largest positive tendency values are dominantly distributed in northern Xinjiang and the smallest negative tendency values are mainly located in southern Gansu, which is consistent with the findings of Shi et al. [32] that the significantly wetter areas are located in the Tianshan Mountains in northern Xinjiang, Northwest China, with a shift of being warm-dry to warm-wet, and the findings of Shen et al. [42] that southern Gansu and eastern Qinghai are significantly drier. In addition, the wetness index increased significantly more than it decreased, which is also consistent with the conclusion that more areas have become wetter than areas that have become drier in China in recent years [42].

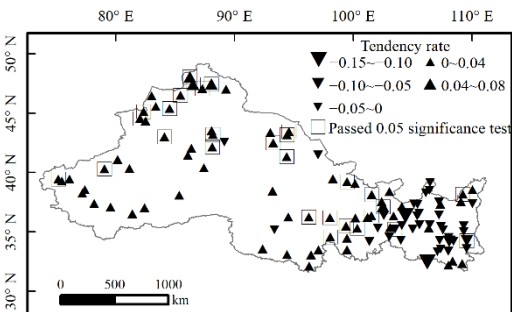

**Figure 3.** Spatial distribution of interannual trends of humidity index in five provinces of Northwest China from 1960 to 2020.

In summary, although the northwest region as a whole has become wetter, there are significant regional differences, a warm-dry trend in the eastern part of the northwest and a warm-wet trend in the western part of the northwest.

*3.2. Mutation Analysis and Period Analysis of the Wetness Index*

3.2.1. Mutation Analysis of the Wetness Index

Using the regime shift detection test to test the interannual and seasonal wetness index in five provinces of Northwest China over the past 61 years, we can see that (Figure 4A,B) the wetness index had two pattern shifts, the pattern shifts occurred in 1986 and 2009, the corresponding pattern shift indexes were 0.24 and 0.19, respectively, with a more obvious sudden change in 1986. This is consistent with the findings of Shi et al. [32], who concluded that the northwest region changed from warm-dry to warm-wet in the late 1980s, and Ma et al. [43], who found that the northwest region became relatively wet after 1985. It is also consistent with the findings that precipitation in the northwest region began to increase in 1986 [44]. In terms of seasons, the spring wetness index underwent a total of two pattern shifts in 1972 and 2012, corresponding to the RSI of 0.41 and 0.22, so the mutation was more prominent in 1972. The summer mutation occurred in 1976 and 2017, corresponding to the RSI of 0.45 and 0.17, so the mutation was most in prominent 1976. The autumn mutation occurred in 1983 and 2014, with RSI of 0.21 and 0.16, so the mutation was more obviously in 1983; the winter wetness index only mutated once in 1988 with RSI of 0.21. Apparently, spring, summer, autumn, and winter have the most significant abrupt changes in the wetness index in 1972, 1976, 1983, and 1988, respectively, with the earliest shift in spring and the strongest response to climate warming, followed by that in summer and then that in autumn and winter.

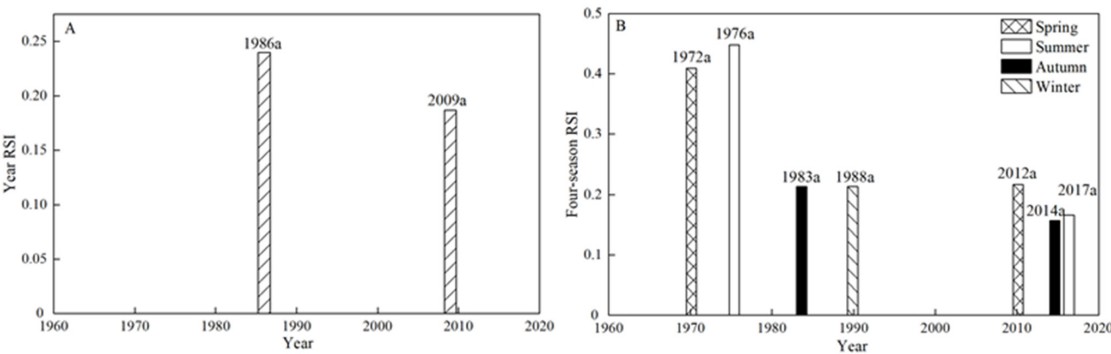

**Figure 4.** Regime shift index of humidity index in five provinces of Northwest China from 1960 to 2020. (**A**) 1960–2020 (**B**) spring, summer, autumn, winter.

### 3.2.2. Period Analysis of the Wetness Index

The results showed that the annual wetness index existed in cycles of 3.61 a, 7.11 a, and 8.83 a in the study area (Figure 5A). Among the seasons, the spring periods were 3.88 a and 4.92 a (Figure 5B), the summer periods were 2.18 a and 2.81 a (Figure 5C), the autumn and winter periods were 3.61 a and 2.10 a, respectively (Figure 5D,E), and all of them passed the 95% confidence level. Manifestly, the wetness index in the study area is characterized by a short period and consistent with the quasi-period of atmospheric circulation 2~4 a and El Niño 2~7 a.

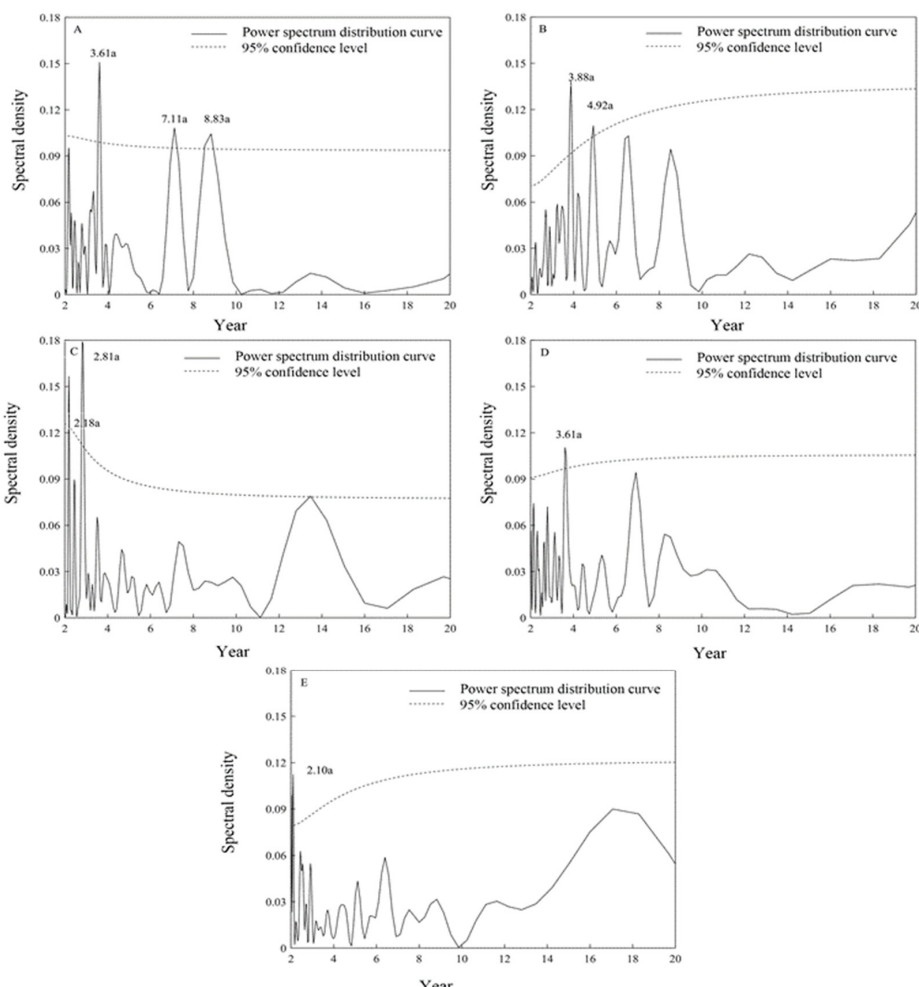

**Figure 5.** Regime shift index of humidity index in five provinces of Northwest China from 1960 to 2020. Note: (**A**) annual cycle (**B**) spring cycle (**C**) summer cycle (**D**) autumn cycle (**E**) winter cycle.

### 3.3. Northwest Dry and Wet Climate Zone

It was shown that the Kriging interpolation method takes into account the altitude of which effect and accuracy are optimal [45]. Therefore, in this study, we use the Kriging interpolation method to obtain the average dry-wet zones distribution map of five provinces of Northwest China from 1960 to 2020 (Figure 6). As Figure 6 already suggests, except for the extreme arid region, the rest of the climatic zones show a clear band-like distribution. The extreme arid region with a wetness index <0.03 is mainly distributed in the southern part of Xinjiang's Qiemo County and Minfeng County. The arid region (0.03~0.2) is mainly distributed in southern Xinjiang, northwestern Gansu, and northern Qinghai. The semi-arid region (0.2~0.5) is mainly distributed in northern Ningxia, the Hexi Corridor, central Qinghai, and northern Xinjiang. The semi-humid region (0.5~0.75) is mainly distributed in southeastern Qinghai, southern Ningxia (0.5~0.75) mainly in southeastern Qinghai, southern Ningxia, northern Shaanxi, and northern Xinjiang. The relatively humid region (0.75~1) is mainly distributed in the Guanzhong Plain of Shaanxi and Tianshui, Pingliang, and Qingyang of Gansu. The humid region (>1) is mainly distributed in the Qinba Mountains of southern Shaanxi and Longnan City of Gansu.

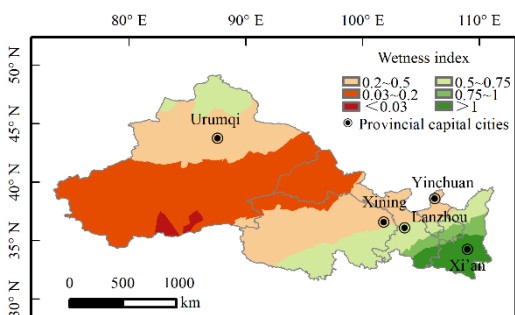

**Figure 6.** Spatial distribution of interannual trends of humidity index in five provinces of Northwest China from 1960 to 2020.

### 3.4. Interdecadal Spatial and Temporal Variability of Dry and Wet Climate Boundaries

In this study, based on the 1960s (1960–1969), we derive and analyze comparatively the fluctuation changes in five dry-wet climate boundaries between each decade (Figure 7). From Figure 7, it can be seen that the boundary of the extreme arid region and the arid region (0.03 boundary line) in the study area started from the southwest corner of Bayangol Mongol Autonomous Prefecture in Xinjiang in the east, passed through the central part of Minfeng and Yutian counties and reached Cele County in the west in the 1960s (Figure 7A). The western section of the boundary retreated to the east in the 1970s and the extreme arid region shrank. The western section of the boundary expanded to the northwest to Hetian and Moyu counties in the 1980s and the area of the extreme arid region increased and reached the peak. The boundary retreated sharply to the south and moved most significantly in the 1990s and the area of the extreme arid region decreased rapidly. By the 2010s the extreme arid region disappeared. It is consistent with previous studies that agree on the shift from warm-dry to warm-wet in the northwest in the late 1980s [32,35].

There are two boundaries (0.2 boundary line) between arid and semi-arid regions, one is located in central Xinjiang and the boundary in central Xinjiang shifted southward as a whole, the other is located in Qinghai and Gansu which are in the southeast of the study area and the boundary shifted northward as a whole (Figure 7B). As is shown in Figure 7B, the boundary in central Xinjiang was distributed roughly along the alignment of the Tianshan Mountains in the 1960s and receded to the southeast in the 1970s, which continued in the 1980s. The boundary shifted to the east and south in the 1990s and continued to shift south in the 2000s and 2010s. The boundary in Qingjiang-Ganzhou starts from the eastern part of the Kunlun Mountains in the west, through the Qaidam Basin to Jiuquan in Gansu. The western part of the line advanced to the northwest in the 1970s

and moved substantially, and the arid area plummeted to the minimum. Compared with the 1970s, the line moved southward to the east in 1980s, but did not reach the position in the 1960s, so the arid area expanded. The boundary advanced to the northwest in the 1990s, and the arid area expanded in comparison with the 1980s. The 2000s and 2010s witnessed boundaries both advancing to the northwest and the arid region area continued to decrease.

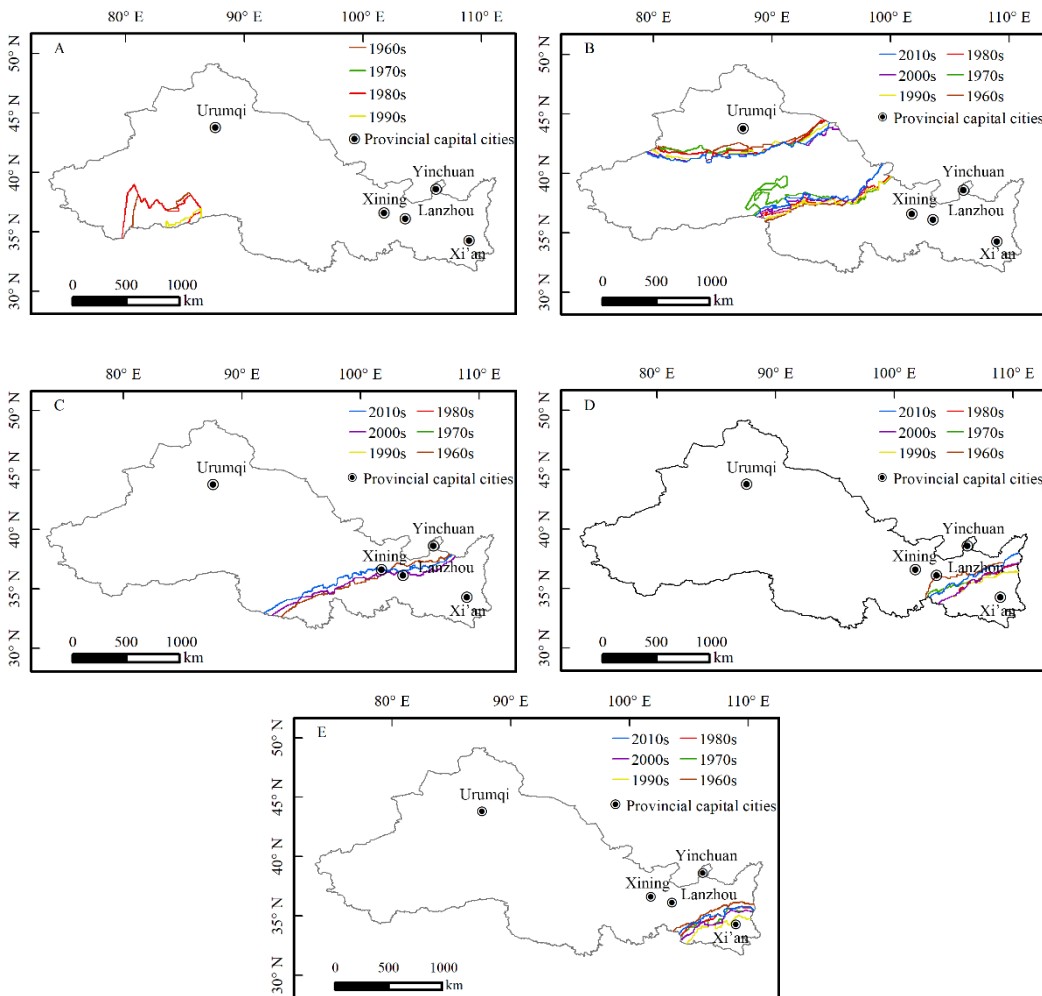

**Figure 7.** Interdecadal fluctuations of each dry and wet climate boundary in five provinces of Northwest China. (**A**): 0.03 boundary (**B**): 0.2 boundary (**C**): 0.5 boundary (**D**): 0.75 boundary (**E**): 1 boundary.

There are also two semi-arid and semi-humid boundaries (0.5 boundary width), one of which is located in the northern part of northern Xinjiang and the other of which in the southeastern part of the study area (Figure 7C). In the past 61 years, the boundary in the northern part of northern Xinjiang has shifted southward and in the southeastern part has shifted northwestward as a whole. The boundary in the northern part of northern Xinjiang was located near the Altay Mountains north of the Junggar Basin in the 1960s but disappeared in the 1970s, indicating that the western part of northwestern Xinjiang was in a drier stage in the 1970s and had not yet embarked on the stage of warming and humidification. The boundary shifted southward in the 1980s in comparison with 1960s, the area of semi-humid region expanded, indicating that northwestern Xinjiang began to be warm and wet. This is consistent with the fact that the precipitation in the high mountainous areas such as Tianshan and Altay Mountains increased significantly in the 1980s and 1990s [32]. The line continued to move southward in the 2000s and 2010s and the area of the semi-humid region in the northern part of Xinjiang continues to increase,

indicating that the western part of Northwest China is still at present in the wetter stage. The other southeastern 0.5 boundary started from the southeast corner of Qinghai in the west in the 1960s, and passes through Xining and Lanzhou to the central part of Ningxia. The western section of the boundary shifted to the northwest in the 1970s, and the eastern section shifted to the southeast to the northernmost part of Shaanxi. The western section of the boundary shifted southward in the 1980s compared with that of 1970s, the western and middle sections shifted to the northwest in comparison with the 1960s, the eastern section shifted to the southeast. The line continued to shift to the southeast in the 1990s, it shifted to the northwest in the 2000s and 2010s, and showed a northward trend in the 2010s relative to the 1960s with the largest movement and the area of the southeastern part of the study area reached the largest semi-humid zone.

In the past 61 years, the boundary of the semi-humid region and the relatively humid region (the 0.75 boundary line) has shifted to the southeast as a whole (Figure 7D), among which, the boundary shifted from Gannan Tibetan Autonomous Prefecture in Gansu through Lanzhou and southern Ningxia to Yulin in Shaanxi in the 1960s. It shifted to the southeast as a whole with the eastern section moving to Yan'an in Shaanxi in the 1970s. It continued to shift southward with the western section shifting southward to Longnan in Gansu and no significant change happened in the eastern section in the 1980s. The boundary in the 1990s and 2000s moved to the greatest extent compared with that in the 1960s and it was of the same in the southernmost position. The line shifted northward as a whole in the 2010s and the eastern section almost coincided with that in the 1960s but the western section did not reach the position in the 1960s.

The boundary of (0.1 boundary width) the relatively humid region and the humid region shifted southward to the east during 1960–2020 (Figure 7E). The boundary started from Longnan in Gansu in the west, went through Tianshui, Pingliang, Qingyang to Yan'an in Shaanxi in the 1960s. The line shifted southward as a whole in the 1970s, the area of the humid region shrank and the eastern part of the northwest began to tend to be warm-dry. The western section of the boundary remained basically unchanged in the 1980s compared with that the 1970s, while the central and eastern sections continued to move southward. The line shifted southward to the greatest extent in the 1990s, the boundary fluctuated from Longnan, Gansu to Baoji, Shaanxi to Xianyang to Tongchuan to Weinan, the area of the humid region decreased rapidly to the minimum value, indicating that the degree of aridification in eastern Northwest China accelerated and reached its peak in the 1990s. This line began to move northward in the 2000s, which basically coincided with the line in the 1970s, and the line in the 2010s almost coincided with that of the 1980s. The area of humid zone has expanded, indicating that the degree of warm-dry in the eastern part of Northwest China has slowed down. This is consistent with the study's finding that precipitation has increased, and warming has slowed in the eastern northwest since the 20th century [46].

To better reflect the fluctuation of dry-wet climate boundaries, we calculated the area of different climate zones in different eras (Table 3) by ArcGis extraction so as to compare the changes in dry-wet zones in different eras. Overall, in the past 61 years, except for the semi-humid region, whose area increased by $26.88 \times 10^4$ km$^2$ (84.11%), the area of all other climatic zones showed a decreasing trend, compared with the 1960s, the extreme arid region would have disappeared by 2010s. The arid region and semi-arid region decreased by about $8.51 \times 10^4$ km$^2$ and $4.68 \times 10^4$ km$^2$ with a total diminution of 5.64%. The area of the relatively humid region and the humid region decreased by $1.13 \times 10^4$ km$^2$ and $2.28 \times 10^4$ km$^2$, respectively, with a total diminution of 12.08%. The area of the extreme arid region inversely increased to the maximum in the 1980s after which they both decreased rapidly. The area of the arid region first decreased in the 1960s and continue to increase in the 1980s and reached the peak in the 1990s after which it decreased sharply, indicating that the transition to warm-wet began only at the end of the 1980s in western Northwest China. The area of the semi-arid region only increased to the vertex in the 1970s after which they both decreased. The area of the semi-humid

region has increased significantly since 1960s, the area of relatively humid region started to decrease since 1960s and dwindled to the lowest in the1980s and experienced the process of expansion–shrinkage–expansion from 1990s to 2010s. The area of humid region fluctuated until the 2000s and started to rise continuously, indicating that the trend of warming and drying in eastern Northwest China has decelerated. The overall area of the three arid climate zones shrank by about $23.47 \times 10^4$ km$^2$ with a decrease of 9.61%, the overall area of the three humid zones expanded by about $23.47 \times 10^4$ km$^2$ with an increase of 39.01%. Evidently, the overall climate of five provinces of Northwest China tends to warm and wet.

**Table 3.** The decadal difference of different climate areas in five provinces of Northwest China.

| Era | Arid Climate Zone | | | | Humid Climate Zone | | | |
|---|---|---|---|---|---|---|---|---|
| | Area of Extreme Arid Zone ($\times 10^4$ km$^2$) | Area of Arid Zone ($\times 10^4$ km$^2$) | Area of Semi-Arid Zone ($\times 10^4$ km$^2$) | Total Area ($\times 10^4$ km$^2$) | Area of Semi-Humid Zone ($\times 10^4$ km$^2$) | Area of Relatively Humid Zone ($\times 10^4$ km$^2$) | Area of Humid Zone ($\times 10^4$ km$^2$) | Total Area ($\times 10^4$ km$^2$) |
| 1960s | 10.2855 | 130.6363 | 103.2051 | 244.1269 | 31.9599 | 10.7316 | 17.4815 | 60.173 |
| 1970s | 9.1238 | 119.0654 | 117.847 | 246.0362 | 35.9854 | 9.1106 | 13.1677 | 58.2657 |
| 1980s | 12.2963 | 121.7662 | 114.0093 | 248.0718 | 36.401 | 5.3684 | 14.4586 | 56.228 |
| 1990s | 0.2974 | 131.6722 | 105.451 | 237.4206 | 48.5137 | 8.7329 | 9.6328 | 66.8794 |
| 2000s | 0.0155 | 123.8233 | 104.6986 | 228.5374 | 55.8629 | 7.6943 | 12.2054 | 75.7626 |
| 2010s | 0 | 122.1303 | 98.5245 | 220.6548 | 58.8417 | 9.5998 | 15.2038 | 83.6453 |

In summary, there are differences in interdecadal fluctuations of the area of different climatic zones, and the core features of the dry-wet variability of the climate in five provinces of Northwest China are embodied in the expansion of the semi-humid region and the reduction in the other climatic zones.

## 4. Discussion

This study uses daily meteorological data from 110 meteorological stations to calculate the wetness index and analyzes the spatiotemporal variability characteristic of the wetness index and different dry-wet climate boundaries in five provinces of Northwest China from 1960 to 2020. It was found that the overall climate of five provinces of Northwest China transformed abruptly in 1986, switching from warm-dry to warm-wet but the spatial distribution differed significantly, i.e., the west and central part tended to be warm-wet and the east tended to be warm-dry, which is consistent with the findings of previous studies [32,35,41]. The seasonal variations, with spring, summer, and early autumn, responded more vehemently to climate warming. Nevertheless, this study finds that the variability of the dry-wet climate zones in five provinces of Northwest China is characterized by the expansion of the semi-humid region and the contraction of the semi-arid region, which is contrary to other researchers' conclusions outlined by Li and Ma [19] that the variability of the dry-wet climate zones in the country is characterized by the shrinkage of the semi-humid region and the expansion of the semi-arid region, indicating that the variability of the dry-wet climate zones has manifest regional differences. Secondly, this study adopts the refined dry-wet climate zoning criteria of Wang et al. [30], analyzes fluctuations of five dry-wet climate boundaries in more details than the existing studies and the time series of dry-wet climate boundaries in this study is longer, reaching 61 years, which better exposes the spatiotemporal variation in dry-wet climate boundaries since the 21st century. The study also concludes an overall shrinkage of 9.61% in the arid climate zones (extreme arid, arid, semi-arid) and an overall expansion of 39.01% in the humid climate zones (semi-humid, relatively humid, humid). In addition, the regime shift detection test was deployed for mutation analysis in this study, which compensates for the shortcomings of the Mann–Kendall mutation test and sliding t-test utilized in previous studies that failed to effectively detect the leap in the later part of the time series. The

Fourier power spectrum is selected for the period analysis, which can desirably reflect the overall characteristics of the whole time period. The shortcomings of this study are that there are more meteorological stations selected in the study area which are not uniformly distributed, moreover, more stations are moved, withdrawn, or missing data, which may affect the accuracy of the results.

## 5. Conclusions

(1) In the past 61 years, the overall trend of the wetness index in five provinces of Northwest China has been increasing, the internal variation shows an augmentation trend of the wetness index in the west and central part of the country where the climate tends to be warm and wet, while the eastern part tends to dwindle where climate tends to be warm and dry.

(2) The annual wetness index in five provinces of Northwest China mutated in 1986 and scaled the heights of the relative wetness with periods of 3.61 a, 7.11 a, and 8.83 a, which are consistent with the cycles of atmospheric circulation and El Niño. There are differences in the timing of mutation in the wetness index in different seasons with dramatic changes in spring, summer, autumn and winter in 1972, 1976, 1983 and 1988, with periods of 3.88 a and 4.92 a, 2.18 a and 2.81 a, 3.61 a and 2.10 a, respectively, which are aligned with the period of atmospheric circulation.

(3) The dry-wet climate boundaries in five provinces of Northwest China fluctuate drastically. The 1960s boundary is taken as the variation baseline, the boundary of extreme arid region and arid region kept moving southward and the extreme arid region disappeared in the 2010s. The boundary between arid and semi-arid regions and semi-arid and semi-humid regions both had two lines, which consistently showed that the northwestern boundary moved southeastward and the southeastern boundary moved northwestward, the area of arid and semi-arid regions retreated substantially, and its area contracted by a total of 5.64% compared with that of 1960s, whereas the area of the semi-humid region expanded by 84.11%. The boundaries of the semi-humid and relatively humid regions and the relatively humid and the humid regions as a whole all moved significantly to the southeast and the area of the relatively humid region and the humid region contracted by a total of 12.08%. In general, the area of arid climate zone dwindled, the area of humid climate zone expanded prominently, and the performance is momentous in the 21st century.

**Author Contributions:** Conceptualization, methodology, M.W. and W.S.; software, M.W., L.L. and W.S.; formal analysis, M.W.; investigation, X.Q. and L.L.; resources, X.Q.; data curation, M.W. and X.Q.; writing—original draft preparation, M.W.; writing—review and editing, P.L.; visualization, L.L.; supervision, P.L.; project administration, P.L.; funding acquisition, P.L. All authors have read and agreed to the published version of the manuscript.

**Funding:** This research was funded by the National Natural Science Foundation of China, grant number 41561080.

**Institutional Review Board Statement:** Not applicable.

**Informed Consent Statement:** Not applicable.

**Data Availability Statement:** Meteorological data covering the analysis period were obtained from https://data.cma.cn (accessed on 3 April 2021).

**Conflicts of Interest:** The authors declare no conflict of interest.

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
