# Peer review of "Spatio-Temporal Characteristics of Dry-Wet Conditions and Boundaries in Five Provinces of Northwest China from 1960 to 2020"

_atmosphere, doi:10.3390/atmos12111499_

Round 1
Reviewer 1 Report
The comments are attached.

Reviewer 2 Report
This study is well written and concise. It is also well documented, incorporating references’ information clearly into the text. However, most references are very old and just two references are recent (e.g. 2019 and 2021). I recommend the authors add more recent references to show that there is no other works in this regard, and to strengthen the importance of this study. Furthermore, the authors need to discuss the results further with the works they themselves present in the introduction. Regarding the text content, minor spelling check is needed to improve the readability. The following modifications are suggested to the authors, in order to increase the quality of the manuscript. Correct the name of section 2.2.2. which is the same as section 2.2.1. In the caption of Figure 2, change EFO to EOF. Again sections 3.2.1 and 3.3 exist twice, so authors need to check and correct them. Finally, in section 3.3 no references were used by the authors to discuss their results.
Reviewer 3 Report
- Title: use Spatio-Temporal and not "Spaito-Temporal"
- Line 133: "the average annual temperature is -11.96-17.61 oC" do you mean the average annual temperature ranges between -11.96 and 17.61 oC?
- lines 133, 134: clarify also the annual sunshine hours (in flat terrain) and the annual average precipitation.
- Equation (1): state that P denotes the daily precipitation; ETo: explain it is the daily reference crop evapotranspiration.
- Lines 66-67: the FAO-Penman Monteith method calculates the reference crop evapotranspiration and not the potential evapotranspiration.
- Equation (2) calculates the reference crop evapotranspiration and not the potential evapotranspiration.
- gamma in equation (2) is the psychrometric parameter;
- Line 191 : should it be XN-1 and not Xn-1?
- Equation (1): clarify that the larger W, the more humid the climate, and vice versa;
- Lines 291, 292: the first line does not have meaning.
- Figure 5, explain parts A, B, C, D, E in the legend.
- Provide a reference for the wetness index, or are you defining it for the first time.
- Provide a reference for equation (3).
Round 2
Reviewer 1 Report
.
Author Response
we sincerely thank the reviewers for their valuable comments and suggestions on our paper. We wish you good health and good work.
Reviewer 3 Report
- Equation (3): define lambda in the complex exponential; "n" must be in the numerator of the complex exponential: revise this equation, see for example Priestly, M.B., Spectral Analysis and Time Series, Academic Press, 1981, equation 7.6.1.
- Line 203: provide a reference for the empirical orthogonal function (EOF);
- Figure 2, right panel: are PC1, PC2, and PC3 the principal components?
- Table 2 needs reformatting.
